# Association of Placental Pathology with Physical and Neuronal Development of Infants: A Narrative Review and Reclassification of the Literature by the Consensus Statement of the Amsterdam Placental Workshop Group

**DOI:** 10.3390/nu16111786

**Published:** 2024-06-06

**Authors:** Chizuko Yaguchi, Megumi Ueda, Yuri Mizuno, Chie Fukuchi, Masako Matsumoto, Naomi Furuta-Isomura, Hiroaki Itoh

**Affiliations:** Department of Obstetrics and Gynecology, Hamamatsu University School of Medicine, Hamamatsu 431-3192, Japan; m.ueda@hama-med.ac.jp (M.U.); yuri-m@hama-med.ac.jp (Y.M.); cfukuchi@hama-med.ac.jp (C.F.); masako.m@hama-med.ac.jp (M.M.); nisomura@hama-med.ac.jp (N.F.-I.); ihiroaki@hama-med.ac.jp (H.I.)

**Keywords:** placenta, nutrients transport, pathology, infants, neural development, physical development

## Abstract

The placenta is the largest fetal organ, which connects the mother to the fetus and supports most aspects of organogenesis through the transport of nutrients and gases. However, further studies are needed to assess placental pathology as a reliable predictor of long-term physical growth or neural development in newborns. The Consensus Statement of the Amsterdam Placental Workshop Group (APWGCS) on the sampling and definition of placental lesions has resulted in diagnostic uniformity in describing the most common pathological lesions of the placenta and contributed to the international standardization of descriptions of placental pathology. In this narrative review, we reclassified descriptions of placental pathology from previously published papers according to the APWGCS criteria and comparatively assessed the relationship with infantile physical and/or neural development. After reclassification and reevaluation, placental pathology of maternal vascular malperfusion, one of the APWGCS criteria, emerged as a promising candidate as a universal predictor of negative infantile neurodevelopmental outcomes, not only in term and preterm deliveries but also in high-risk groups of very low birthweight newborns. However, there are few studies that examined placental pathology according to the full categories of APWGCS and also included low-risk general infants. It is necessary to incorporate the assessment of placental pathology utilizing APWGCS in the design of future birth cohort studies as well as in follow-up investigations of high-risk infants.

## 1. Introduction

The placenta plays a critical role in the regulation of the transport of nutrients from the mother to the fetus and supports most stages of fetal organogenesis [1,2,3]. The deterioration of placental transfer may result in an insufficient supply of nutrients, oxygen, and various bioactive factors, including hormones, immune substances, and protective agents against infection [4].

There is increasing evidence indicating that the condition of the placenta, including its adaptations to the surrounding environment, significantly influences the in utero fetal programming process, affecting both the health and risk of non-communicable diseases long after birth [3,5,6,7,8]. This is termed the Developmental Origin of Health and Disease theory [9,10]. Anna A Penn proposed the concept of ‘neuroplacentology’ and described how placental dysfunction or abruption may program the developing brain for long-term neurological and psychiatric morbidities [11,12,13]. We recently demonstrated that a close correlation has been hypothesized between fetoplacental pathology and atopic dermatitis in female infants [14]. Barker et al. proposed the concept of ‘placental origins of chronic disease’ [15]. More recently, Gardella et al. proposed a new concept of ‘Neuroplacentology’ focusing on the role of the placenta in fetal and neonatal brain development [16], and Kratimenos demonstrated ‘Placental programming of neuropsychiatric disease’ [12]. The placenta represents structural adaptive changes in response to environmental disruption *in utero*. Historically, it has been applied to the evaluation of placental conditions, including malfunction, and is referred to as the ‘memory of a pregnancy’ [4]. Verde et al. demonstrated the association between fetal Doppler and uterine artery blood volume flow in term pregnancies, suggesting a possible contribution of the placenta to fetal outcome [17].

Placental pathology reflects pathophysiological changes and physiological placental adaptation or maladaptation to various environmental factors from both the maternal and fetal sides [4,18]. Several large-scale studies examined the relationship between placental histology and outcomes of newborns in high-risk cases such as severe intrauterine infections, preterm labor, and fetal hypoxia [19,20,21,22]. However, further studies are needed to establish placental pathology as a reliable universal predictor of long-term physical growth or neural development in newborns. We hypothesize that this is because most classifications and descriptions of placental pathological lesions were not universally standardized, and most studies focused on the follow-up of high-risk newborns without comparing them to low-risk control groups.

The placental pathology has been useful information for understanding the pathogenesis of perinatal or neonatal abnormality [23]. However, the variability in the nomenclature used to describe placental lesions made it difficult to understand the correlation between clinical outcome and placental pathology [23]. The Consensus Statement of the Amsterdam Placental Workshop Group (APWGCS) on the sampling and definition of placental lesions [24], published by Khong in 2016, has resulted in diagnostic uniformity in describing the most common pathological lesions of the placenta and contributed to the international standardization of descriptions of placental pathology. Indeed, Slack et al. reviewed the most salient aspect of the emerging literature-based impact by APWGCS [25]. Gardella et al. reported an association between placental pathological findings by APWGCS and neurodevelopmental outcomes at 2 years old in severe growth-restricted fetuses [26]. In a previous study using APWGCS criteria, we analyzed the pathology of 258 whole placentas from singleton pregnancies enrolled in the Hamamatsu Birth Cohort for Mothers and Children [27] and identified maternal vascular malperfusion (MVM) as a significant predictor of lower body weight and deciduitis as a significant predictor of a small ponderal index [28]. Moreover, some pathological findings, such as accelerated villous maturation, were significant predictors of lower Mullen Scales of Early Learning (gross motor, visual reception, fine motor, receptive language, expressive language) at 10, 14, 18, 24, 32, and 40 months [29], whereas others, such as fetal vascular malperfusion (FVM), were significant predictors of higher MSEL composite scores in relation to neurodevelopmental milestones [30]. In the present study, we hypothesized that reevaluating the previous literature by reclassifying their placental pathological statements using APWGCS criteria would provide insight into establishing placental pathology as a clinically useful predictor of infantile development.

The specific aims of the present study were as follows: (1) to search for previous articles describing placental pathology and infantile development, (2) to reclassify their placental pathological statements using APWGCS, and (3) to reevaluate previously reported evidence concerning infantile physical growth and neuronal development for each reclassified APWGCS placental pathological category.

## 2. Materials and Methods

We searched for relevant articles in PubMed using the common keyword of ‘placental pathology’ with the following keywords: ‘neurologic’ or ‘cerebral palsy’ or ‘neurodevelopmental outcome’ or ‘follow up’. We collected 614 articles. We excluded articles that were abstracts, focused solely on fetal and/or neonatal death, were not original submissions, and those without descriptions of newborn outcomes. Following exclusion criteria, 62 articles remained. Next, we excluded articles describing only one category of placental pathologies, such as ‘chorioamnionitis’, and articles that included only high perinatal risk cases without a control group. One article was added in the review process. After these processes, nine articles remained, of which the APWGCS criteria were used in three articles, including ours. We reclassified the pathological findings of the other five articles using APWGCS and reevaluated a total of eight articles (Table 1).

Descriptions of placental pathological findings were classified into categories, with modifications according to our recent study [28], in consideration of the current APWGCS criteria [24]. Classifications included the following: ‘accelerated villous maturation’ (Figure 1A), ‘decidual vasculopathy’ (Figure 1B), ‘distal villous hypoplasia’ (Figure 1C), ‘thrombus’ (Figure 1D), ‘avascular villi’ (Figure 1E), ‘maternal inflammatory response (MIR)’ (Figure 1F), ‘fetal inflammatory response (FIR)’ (Figure 1G), and ‘villitis of unknown etiology (VUE)’ (Figure 1H).

(1) ‘Accelerated villous maturation’ was reclassified if the description indicated increased numbers of placental villi with the focal formation of tight adherent villous clusters [24] typically with syncytial knots and increased perivillous fibrin [31] (Figure 1A); (2) ‘decidual vasculopathy’ was reclassified if the description indicated vascular lesions, including the fibrinoid necrosis of decidual vessels or arthrosis at the basal plate [24] (Figure 1B); (3) ‘distal villous hypoplasia’ was reclassified if the description indicated thin and relatively elongated-appearing villi with increased syncytial knots [24] (Figure 1C); (4) ‘thrombus’ was reclassified if the description indicated localized, protuberant mural lesions composed of proliferating fibroblasts intermixed with fibrin and erythrocytes in the walls of large placental vessels, according to the description by Desa [24,32] (Figure 1D); (5) ‘avascular villi’ was reclassified if the description indicated the total loss of villous capillaries and bland hyaline fibrosis in an older lesion [24] (Figure 1E); (6) ‘MIR’ was reclassified if the description indicated the infiltration of neutrophils into the connective tissues of the chorionic plate and/or amnion basement membrane in the fetal surface of the placenta [4,24,33] (Figure 1F); (7) ‘FIR’ was reclassified if the description indicated the infiltration of neutrophils into umbilical vessels or chorionic plate vessels [4,24,34] (Figure 1G); and (8) ‘VUE’ was reclassified if the description indicated lymphohistiocytic inflammation predominantly localized to the villous stroma of terminal villi despite the absence of clinical symptoms of apparent infection in mothers or infants [4,24,35] (Figure 1H).

According to the APWGCS [24], ‘MVM’ was reclassified if the description indicated ‘accelerated villous maturation’, ‘decidual arteriopathy’, ‘distal villous hypoplasia’, ‘infarction’, and/or ‘other findings’ such as ‘intervillous thrombosis’. Furthermore, ‘FVM’ was reclassified if the description indicated ‘FVM’, ‘avascular villi’, ‘fetal thrombosis’, and/or ‘villous stromal vascular karyorrhexis’, with consideration of intermittent cord obstruction.

In the present study, each of the descriptions of pathological findings was reclassified as positive or negative by three researchers who specialize in placental pathology (Drs. Chizuko Yaguchi, Naomi Furuta-Isomura, and Megumi Ueda) [30,36,37].

## 3. Results

### 3.1. Articles Reclassified Using APWGCS

After searching PubMed using keywords related to placental pathology and infantile outcomes, we selected relevant articles that met the inclusion criteria (Table 1). APWGCS criteria were used in three of eight articles. Two studies included a general low-risk general population that was enrolled in the birth cohort study early in pregnancy [28,30]. One study included placentas from infants who were small for gestational age (SGA) at term and late preterm (>34 weeks) [38]. Three studies selected placentas from early preterm deliveries [39,40,41], and two studies included placentas from early preterm deliveries and/or infants with very low birthweight (<1500 g) [26,42]. One study included placentas from newborn infants who were Hypoxic– Ischemic Encephalopathy at term and late preterm (≥36 weeks) [43].

**Table 1 nutrients-16-01786-t001:** Articles reevaluated in the present study.

First Author (Publication Year)	Country	Study Period	Sample Size	Selection Criteria	Postnatal Evaluation Period	Reference of Placental Pathology
Yaguchi C et al. (2018) [28]	Japan	2007–2011	258	Singleton pregnancy. General population between 29 and 42 wks	18 m.o.	APWGCS
Ueda M et al. (2020) [30]	Japan	2007–2011	258	Singleton pregnancy. General population between 29 and 42 wks	40 m.o.	APWGCS
Parra-Saavedra M et al. (2014) [38]	Spain	2010–2012	83	SGA infants delivered at >34 wks	2 y.o.	International Federation of Placenta Associations and Society for Pediatric Pathology (perinatal section)
Chalak L, et al. (2021) [43]	USA	N/A	321	newborn infants with hypoxic–ischemic encephalopathy (HIE), Delivered at ≥36 wks	2 y.o.	APWGCS
Vinnars MT et al. (2015) [39]	Sweden	2004–2007	139	Delivered at 22–26 wks	2.5 y.o.	N/A
Mir IN et al. (2020) [40]	USA	2009–2012	241	Delivered at <29 wks	2 y.o.	Modified Redline classification
Roescher AM et al. (2011) [41]	The Netherlands	N/A	40	Singleton pregnancy. Delivered at 25.4–31.7 wks	24 h	Royal College of Obstetricians and Gynaecologists, Royal College of Pathologists, and College of American Pathologists
van Vliet EOG et al. (2012) [42]	The Netherlands	2001–2003	72	Singleton pregnancy. Delivered at <32 wks or birthweight of <1500 g	2 y.o./7 y.o.	N/A
Gardella B, et al. (2021) [26]	Italy	2007–2015	20	Singleton pregnancy. FGR delivered at ≤34 wks and birthweight of ≤1500 g	2 y.o.	APWGCS

wks: weeks of gestation. y.o.: years old. m.o.: months old. APWGCS: The Consensus Statement of the Amsterdam Placental Workshop Group. SGA: small for gestational age.

### 3.2. Maternal Vascular Malperfusion (MVM) and Infantile Outcomes

All eight studies included pathological findings related to MVM; thus, they were reclassified as MVM (Table 2). In the general population enrolled in regional birth cohort studies, negative effects were reported on infantile body weight [28] and on MSEL composite scores in neurodevelopment [30]. Parra-Saavedra et al. reported negative effects on Bayley neurodevelopmental scores (Cognitive, Language, Motor) among placentas from late preterm and term (>34 weeks) SGA infants, compared to those with MIR (chorioamnionitis) [38]. However, Parra-Saavedra et al. classified cases positive for MVM or FVM as placental under-perfusion (PUP)-positive. Chalak et al. mixed MVM, FVM, and VUE together as a group of placental chronic abnormalities, having a negative association with base deficit within the first hour of birth newborns with hypoxic–ischemic encephalopathy (HIE), delivered at ≥36 weeks [43]. Of the three studies of placentas from simple, early preterm deliveries, one study reported a negative association with cerebral palsy (CP) at 2.5 y.o. [39], whereas the other two studies reported no specific association with infantile outcomes [40,41] (Table 2). As for the mixed groups of placentas from both early preterm deliveries and/or very low birthweight newborns, one study reported a negative association with mental development at 2 y.o. but not at 7 y.o. [42], whereas another reported no specific association [26].

### 3.3. Fetal Vascular Malperfusion (FVM) and Infantile Outcomes

Seven studies had pathological descriptions related to FVM, whereas one study had no description (Table 3). As for the general population enrolled in the regional birth cohort study, positive associations were reported as MSEL composite scores in neurodevelopment [30] but not body weight or composition [28]. Regarding placentas from SGA infants delivered late preterm and term (>34 weeks), Parra-Saavedra et al. reported a negative association with Bayley neurodevelopmental scores compared to those with MIR (chorioamnionitis) [38]. However, Parra-Saavedra et al. classified cases positive for MVM or FVM as PUP-positive. Chalak et al. mixed MVM, FVM, and VUE together as a group of placental chronic abnormalities, having a negative association with base deficit within the first hour of birth newborns with HIE, delivered at ≥36 weeks [43]. Among the three studies of placentas from simple, early preterm deliveries, one study reported a negative association with the Score for Neonatal Acute Physiology Perinatal Extension within 24 h after birth [41], whereas two studies reported no association with infantile outcomes [39,40]. Among two mixed groups of placentas from both early preterm deliveries and/or very low birthweight newborns, one study reported a negative effect on neurodevelopment at 2 y.o. [26], whereas the other study did not include a description [42].

### 3.4. Maternal Inflammatory Response (MIR) and Infantile Outcomes

Seven articles included pathological descriptions related to MIR; however, one included no description (Table 4). In the general population enrolled in the regional birth cohort study, no specific association with infantile outcomes was observed [28,30]. Parra-Saavedra et al. reported no description regarding placentas from SGA infants delivered late preterm and term (>34 weeks) [38]. Moreover, three studies of placentas from simple, early preterm deliveries reported no specific associations [39,40,41]. Among two mixed groups of placentas from both early preterm deliveries and/or very low birthweight newborns, one reported a relatively positive association with neurodevelopment at 2 y.o. compared to MIR [42], whereas the other reported no specific association [26].

### 3.5. Fetal Inflammatory Response (FIR) and Infantile Outcomes

Five studies included pathological descriptions related to FIR, whereas three articles had no description (Table 5). Concerning the general population enrolled in a regional birth cohort study, no specific association with infantile outcome was observed [28,30]. As for placentas from SGA infants delivered late preterm and term (>34 weeks), there was no description [38]. Among three studies of placentas from simple, early preterm deliveries, only one study found no specific associations [40]. Among two studies of mixed groups of placentas from both early preterm deliveries and/or very low birthweight newborns, one study reported no description [42], and another reported no specific association [26].

### 3.6. Villitis of Unknown Etiology (VUE) and Infantile Outcomes

Seven studies had pathological descriptions related to VUE, and two had no description (Table 6). Chalak et al. mixed MVM, FVM, and VUE together as a group of placental chronic abnormalities, having a negative association with base deficit within the first hour of birth newborns with HIE, delivered at ≥36 weeks [43]. One study reported no cases [40], whereas all of the other studies reported no specific association with infantile outcomes (Table 6).

### 3.7. Numbers of Non-Specific Placental Pathological Lesions and Infantile Outcomes

Mir et al. reported that the number of non-specific placental pathological lesions had a negative association with neurodevelopmental impairment (NDI) and/or bronchopulmonary dysplasia (BPD) [40] (Table 1, Table 2, Table 3, Table 4, Table 5 and Table 6). However, we were unable to fully reevaluate their model using APWGPS because a detailed dataset was not available.

## 4. Discussion

The placenta supplies nutrients and oxygen to the growing fetus throughout pregnancy [2,3]. Malfunction of the placenta may affect fetal growth and development, potentially leading to negative effects on infantile physical and/or neuronal development [15]. Placental pathology represents not only physiological but also pathological changes in the placenta [4]. APWGCS [24] has universally standardized the description of placental pathology; thus, in the present study, we reclassified the placental pathology of the previous literature using APWGCS criteria and reevaluated the relationship between placental pathology and infantile outcomes.

Placental pathological findings change as pregnancy progresses; therefore, pathological findings are different between preterm and term births [4,18]. Moreover, fetal growth restriction is closely associated with placental insufficiency in nutrient supply, and placental pathological findings exhibit dynamic changes depending on the status of placental dysfunction. Their findings suggest that placental pathological findings in fetal growth restriction cases are different from those of normal fetal growth. As many of the studies that were reclassified in the present study selected placentas from preterm deliveries and/or small fetuses, further studies with similar selection criteria are necessary.

Two studies analyzed placentas from the general population within a birth cohort [28,30], and one study analyzed small newborns delivered at term and late preterm (>34 weeks) [38] (Table 1). Since the majority of placentas from these three studies were delivered at term, we discussed these three studies collectively as a near-term group. All three studies reported that MVM had a negative effect on neurodevelopment and body weight (Table 2). On the other hand, FVM had a positive association with neurodevelopment in one study but not in the other studies (Table 3). Although it is difficult to draw a single conclusion concerning the relationship between FVM and infantile outcomes in this near-term group, the former is the only report of a positive association with neurodevelopment among all eight studies (Table 3). Further studies are necessary to clarify whether FVM could be a potential biomarker for better infantile neurodevelopment in newborns delivered at near term. One study mixed MVM, FVM, and VUE together as a group of placental chronic abnormalities, having a negative association with base deficit within the first hour of birth of newborns with HIE delivered at ≥36 weeks [43]. Among the three studies, two described MIR (Table 4) and FIR (Table 5), both of which reported no specific associations with infantile outcomes. This suggests that the effect of acute inflammation is transient and has a minor effect on infantile outcomes. Furthermore, VUE had no association with infantile outcomes (Table 6).

Three studies selected placentas from early preterm deliveries (<26 weeks, <29 weeks, and <31.7 weeks) [39,40,41] (Table 1); thus, we discussed these three studies collectively as an early preterm group. Mir et al. reported that multiple pathological lesions had a negative effect on NDI and/or BPD [40]; however, we were unable to clearly reclassify their scoring using APWGCS criteria. Further studies are necessary to evaluate if the scoring of positive placental pathological findings by APWGCS criteria is correlated with infantile outcomes. The main pathological criteria of APWGCS, such as MVM and FVM, represent groups of pathological lesions associated with specific pathophysiological characteristics of placental dysfunction; therefore, the concept of non-specific summation of positive pathological findings by Mir et al. may not match APWGCS. Further large-scale studies are necessary to introduce this concept to the interpretation of APWGCS. Thus, in the present study, we mainly discussed the other two studies.

Regarding CP, Vinnars et al. reported a negative effect from MVM (Table 2) but found no association with FVM (Table 3) [39]. They reported placentas from the earliest weeks of gestation (22–26 weeks) among the articles reevaluated (Table 1). The newborns constituted an extremely high-risk population, potentially linked to the observed negative association with CP, a severe complication within the spectrum of neurodevelopmental disorders. On the other hand, Rochester et al. reported no association between MVM and the outcomes of newborns 24 h after birth (Table 2) but found that FVM had a negative association (Table 3) [41]. Thus, there may be a negative trajectory in placentas from early preterm deliveries due to MVM and/or FVM toward infantile outcomes; however, this has not been specifically confirmed. There were no associations with MIR (Table 4) and no description of FIR (Table 5). Regarding VUE, one study reported no association, and another described no cases (Table 6). However, the number of placentas positive for VUE was small. Spinillo et al. carried out a systemic review of the articles and pointed out a possible association between FVM and neurodevelopmental injuries in term neonates [44]. Large-scale studies are necessary to evaluate its association with infantile outcomes.

Two studies selected placentas from early preterm deliveries and/or very low birthweight infants (<1500 g) [26] (Table 1). van Vliet et al. compared the Mental Developmental Index (MDI) between MVM and MIR and concluded that MVM had a comparatively negative association with MDI [42]. However, they did not find that MVM was a negative predictor of neurodevelopment nor that MIR was a positive one among entire placentas from early preterm deliveries and/or very low birthweight infants. Gardella et al. reported a negative association with neurodevelopment from placentas with FVM but no association with MVM, MIR, FIR, and VUE [26]. These studies suggest the negative effect of FVM; however, further studies are needed. The pathological description of FVM was considered insufficient in three of five studies of placentas at early preterm delivery (Table 3), as FVM is a recent pathological concept demonstrated by APWGCS [24]. Further investigations incorporating appropriate pathological assessment of FVM are necessary to assess whether it is a predictor of infantile outcomes.

There were some limitations of the reclassification by APWGCS in the present study, such as (1) insufficiency or lack of the descriptions of gross appearance, (2) insufficient pathological descriptions of the relatively new concept of FVM, (3) scoring by mixing several APWGCS criteria together, (4) a possibility of selection vias, etc. We could not fully overcome the limitation.

Collectively, placental pathology of MVM, one of the APWGCS criteria, had a negative association with infantile neurodevelopment in the placentas in a general population [30], term and late preterm deliveries (>34 weeks) [38], and early preterm deliveries (<26 weeks) [38]. Two studies included early preterm deliveries and/or very low birthweight (<1500 g), of which a negative association between MVM and infantile neurodevelopment compared to MIR was found in one study [42]. Taken together, MVM may be a universal predictor of a negative association with infantile neurodevelopmental outcomes, suggesting that it may be a biomarker for targeting early intervention as precision medicine of pediatric neurodevelopmental care. Further studies are necessary to confirm this speculation. Few studies examined placental pathology according to the full categories of APWGCS and also included not only placentas from high-risk newborns but also low-risk general ones. Our findings suggest that it is necessary to introduce the assessment of placental pathology using APWGCS when planning new birth cohort studies of the general population as well as follow-up studies of high-risk infants.

## 5. Conclusions

Placental pathology of maternal vascular malperfusion (MVM), one of the APWGPS criteria, may be a possible candidate as a predictor of negative infantile neurodevelopmental outcomes. It is interesting to speculate that MVM may be a promising biomarker candidate for early preventive intervention to improve neurodevelopment. Further studies are necessary to prove the speculation.

## Figures and Tables

**Figure 1 nutrients-16-01786-f001:**
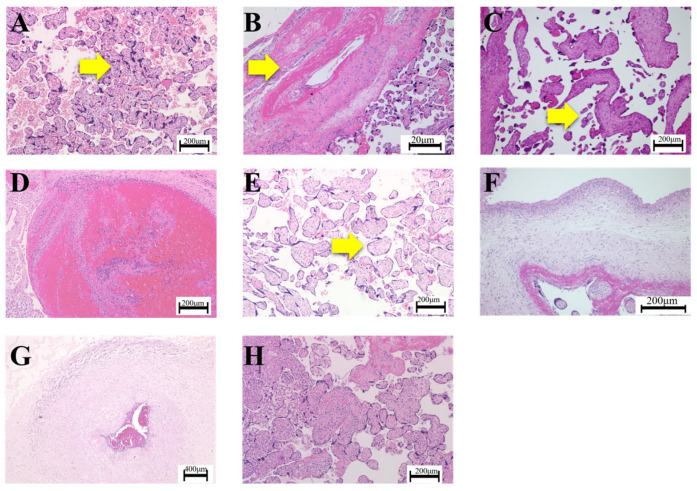
Representative pathological findings by HE staining of placentas in accordance with the categories of the Consensus Statement of the Amsterdam Placental Workshop Group (APWGCS). (**A**) ‘Accelerated villous maturation’; the yellow arrow indicates increased placental villi with the focal formation of tight adherent villous clusters with syncytial knots. (**B**) ‘Decidual vasculopathy’; the yellow arrow indicates the fibrinoid necrosis of decidual vessels. (**C**) ‘Distal villous hypoplasia’; the yellow arrow indicates thin and elongated villi surrounding the stem villi. (**D**) ‘Thrombus’. (**E**) ‘Avascular villi’; the yellow arrow indicates a villus with hyalinized stroma, which is devoid of vessels. (**F**) ‘MIR’; the infiltration of neutrophils into the chorionic plate. (**G**) ‘FIR’; the infiltration of neutrophils into the umbilical artery. (**H**) ‘VUE’; the yellow arrow indicates lymphohistiocytic inflammation predominantly in the stroma of the terminal villi.

**Table 2 nutrients-16-01786-t002:** MVM and infantile outcomes.

First Author (Publication Year)	Pathological Descriptions for Reclassifying as ‘MVM’ Using APWGCS Criteria						Selection Criteria	Postnatal Evaluation Period	Outcomes
	MVM	Accelerated Villous Maturation	Decidual Arteriopathy	Distal Villous Hypoplasia	Infarction	Other			
Yaguchi C et al. (2018) [28]	AD	AD	AD				Singleton pregnancy. General population between 29 and 42 wks	18 m.o.	Negative impact on body weight
Ueda M et al. (2020) [30]	AD	AD	AD				Singleton pregnancy. General population between 29 and 42 wks	40 m.o.	Negative impact on Mullen Scale of Early Learning composite scores in neurodevelopment
Parra-Saavedra M et al. (2014) [38]	Placental under-perfusion	Excessive intervillous fibrin syncytial knots involving terminal villi and villous agglutination (>50%)	Acute atherosis and mural hypertrophy	Undergrowth/distal villous hypoplasia	Intervillous fibrin deposition and villous infarcts	Specific vascular alterations indicative of maternal vascular loss of integrity include arterial rupture (abruption placenta) and venous rupture (acute and chronic marginal abruption). Maldevelopmental vascular alterations include chorioangioma and chorioangiosis	SGA infants delivered at >34 wks	2 y.o.	# Negative impact on Bayley neurodevelopmental scores (Cognitive, Language, Motor) compared to MIR (chorioamnionitis)
Chalak L, et al. (2021) [43]	AD						Newborn infants with HIE, delivered at ≥36 wks	2 y.o.	^$^ Negative impact on base deficit within the first hour of birth
Vinnars MT et al. (2015) [39]		Abnormal villous maturation	Decidual arteriopathy		Placental infarction, placental abruption	Intervillous thrombosis	Delivered at 22–26 wks	2.5 y.o.	Negative impact on the occurrence of Cerebral palsy
Mir IN, et al. (2020) [40]			Severe maternal decidual vasculopathy	Distal villous hypoplasia	Infarcts		Delivered at <29 wks	2 y.o.	No specific association, although non-specific multiple pathological lesions had a negative association with neurodevelopmental impairment and/or bronchopulmonary dysplasia
Roescher AM et al. (2011) [41]	Maternal vascular under-perfusion						Singleton pregnancy. Delivered at 25.4–31.7 wks	24 h	No association
van Vliet EOG et al. (2012) [42]	Placental under-perfusion	Increased syncytial knots, villous agglutination		Distal villous hypoplasia		Intervillous fibrin	Singleton pregnancy. Delivered at <32 wks or birthweight of <1500 g	2 y.o./7 y.o.	Negative association with Mental Developmental Index score at 2 y.o. but not at 7 y.o., compared to MIR
Gardella B et al. (2021) [26]	AD	AD	AD	AD	AD	Arterial or venous abruption	Singleton pregnancy. FGR delivered at ≤34 wks and birthweight of ≤1500 g	2 y.o.	No association

wks: weeks of gestation. y.o.: years old. m.o.: months old. AD: appropriate description for APWGCS. SGA: small for gestational age. #: if either MVM or FVM was positive, it was classified as placental under-perfusion (PUP)-positive. ^$^: MVM, FVM, and VUE were mixed together as a group of chronic placental abnormalities.

**Table 3 nutrients-16-01786-t003:** FVM and infantile outcomes.

First Author (Publication Year)	Pathological Descriptions for Reclassifying as FVM Using APWGCS Criteria					Selection Criteria	Postnatal Evaluation Period	Outcomes
	FVM	Avascular Villi	Fetal Thrombosis	Villous Stromal Vascular Karyorrhexis	Consideration			
Yaguchi C et al. (2018) [28]	AD	AD	AD			Singleton pregnancy. General population between 29 and 42 wks	18 m.o.	No association
Ueda M et al. (2020) [30]	AD	AD	AD			Singleton pregnancy. General population between 29 and 42 wks	40 m.o.	Positive impact on Mullen Scale of Early Learning composite scores in neurodevelopment
Parra-Saavedra M et al. (2014) [38]	Fetal vascular supply	Villous avascularity	Thrombosis of chorionic plate and stem villous channels			SGA infants delivered at >34 wks	2 y.o.	# Negative association with Bayley neurodevelopmental scores (Cognitive, Language, Motor) compared to MIR (chorioamnionitis)
Chalak L, et al. (2021) [43]	AD					Newborn infants with HIE, delivered at ≥36 wks	2 y.o.	^$^ Negative impact on base deficit within the first hour of birth
Vinnars MT et al. (2015) [39]			Fetal thrombosis			Delivered at 22–26 wks	2.5 y.o.	No association
Mir IN et al. (2020) [40]	Fetal thrombotic vasculopathy	AD	Segmental or complete obstruction	Villous stromal vascular karyorrhexis		Delivered at <29 wks	2 y.o.	No specific association, although non-specific multiple pathological lesions had negative association with neurodevelopmental impairment and/or bronchopulmonary dysplasia
Roescher AM et al. (2011) [41]			Fetal thrombotic vasculopathy			Singleton pregnancy. Delivered at 25.4–31.7 wks	24 h	Negative impact on the Score of Neonatal Acute Physiology Perinatal Extension (SNAPPE)
van Vliet EOG et al. (2012) [42]	N/A					Singleton pregnancy. Delivered at <32 wks or birthweight of <1500 g	2 y.o./7 y.o.	N/A
Gardella B et al. (2021) [26]	AD	AD	Intramural fibrin of large placental vessels		Intermittent cord obstruction, ectasia	Singleton pregnancy. FGR delivered at ≤34 wks or birthweight of ≤1500 g	2 y.o.	Negative impact on neurodevelopment

wks: weeks of gestation. y.o.: years old. m.o.: months old. AD: appropriate description for APWGCS. SGA: small for gestational age. #: if either MVM or FVM was positive, it was classified as placental under-perfusion (PUP)-positive. ^$^: MVM, FVM, and VUE were mixed together as a group of chronic placental abnormalities.

**Table 4 nutrients-16-01786-t004:** MIR and outcome of the offspring.

First Author (Publication Year)	Pathological Descriptions for Reclassifying as ‘MIR’ of APWGCS	Selection Criteria	Postnatal Evaluation Period	Outcomes
Yaguchi C et al. (2018) [28]	AD	Singleton pregnancy. General population between 29 and 42 wks	18 m.o.	No association
Ueda M et al. (2020) [30]	AD	Singleton pregnancy. General population between 29 and 42 wks	40 m.o.	No association
Parra-Saavedra M et al. (2014) [38]	N/A	SGA infants delivered at >34 wks	2 y.o.	N/A
Chalak L, et al. (2021) [43]	AD	Newborn infants with HIE, delivered at ≥36 wks	2 y.o.	N/A
Vinnars MT et al. (2015) [39]	Acute chorioamnionitis	Delivered at 22–26 wks	2.5 y.o.	No association
Mir IN et al. (2020) [40]	Chorioamnionitis	Delivered at <29 wks	2 y.o.	No specific association, although non-specific multiple pathological lesions had negative association with neurodevelopmental impairment and/or bronchopulmonary dysplasia
Roescher AM et al. (2011) [41]	Ascending intrauterine infection	Singleton pregnancy. Delivered at 25.4–31.7 wks	24 h	No association
van Vliet EOG et al. (2012) [42]	Chorioamnionitis	Singleton pregnancy. Delivered at <32 wks or birthweight of <1500 g	2 y.o./7 y.o.	Positive association with Bayley neurodevelopmental scores (Cognitive, Language, Motor) compared to MVM
Gardella B et al. (2021) [26]	Acute chorioamnionitis (stage and grade)	Singleton pregnancy. FGR delivered at ≤34 wks and birthweight of ≤1500 g	2 y.o.	No association

wks: weeks of gestation. y.o.: years old. m.o.: months old. AD: appropriate description for APWGCS. SGA: small for gestational age.

**Table 5 nutrients-16-01786-t005:** FIR and infantile outcomes.

First Author (Publication Year)	Pathological Descriptions for Reclassifying as ‘FIR’ Using APWGCS Criteria	Selection Criteria	Postnatal Evaluation Period	Outcomes
Yaguchi C et al. (2018) [28]	AD	Singleton pregnancy. General population between 29 and 42 wks	18 m.o.	No association
Ueda M et al. (2020) [30]	AD	Singleton pregnancy. General population between 29 and 42 wks	40 m.o.	Positive association with Mullen Scale of Early Learning composite scores in neurodevelopment
Parra-Saavedra M et al. (2014) [38]	N/A	SGA infants delivered at >34 wks	2 y.o.	N/A
Chalak L, et al. (2021) [43]	AD	Newborn infants with HIE, delivered at ≥36 wks	2 y.o.	N/A
Vinnars MT et al. (2015) [39]	N/A	Delivered at 22–26 wks	2.5 y.o.	N/A
Mir IN et al. (2020) [40]	Vasculitis in the umbilical vessels and/or chorionic plate vessels	Delivered at <29 wks	2 y.o.	No specific association, although non-specific multiple pathological lesions had a negative association with neurodevelopmental impairment and/or bronchopulmonary dysplasia
Roescher AM et al. (2011) [41]	N/A	Singleton pregnancy. Delivered at 25.4–31.7 wks	24 h	N/A
van Vliet EOG et al. (2012) [42]	Chorioamnionitis with an additional fetal response	Singleton pregnancy. Delivered at <32 wks or birthweight of <1500 g	2 y.o./7 y.o.	N/A
Gardella B et al. (2021) [26]	Fetal inflammatory acute response (stage and grade)	Singleton pregnancy. FGR delivered at ≤34 wks and birthweight of ≤1500 g	2 y.o.	No association

wks: weeks of gestation. y.o.: years old. m.o.: months old. AD: appropriate description for APWGCS. SGA: small for gestational age.

**Table 6 nutrients-16-01786-t006:** VUE and infantile outcomes.

First Author (Publication Year)	Pathological Descriptions for Reclassifying as VUE Using APWGCS Criteria	Selection Criteria	Postnatal Evaluation Period	Outcomes
Yaguchi C et al. (2018) [28]	AD	Singleton pregnancy. General population between 29 and 42 wks	18 m.o.	No association
Ueda M et al. (2020) [30]	AD	Singleton pregnancy. General population between 29 and 42 wks	40 m.o.	No association
Parra-Saavedra M et al. (2014) [38]	N/A	SGA infants delivered at >34 wks	2 y.o.	N/A
Chalak L, et al. (2021) [43]	AD	Newborn infants with HIE, delivered at ≥36 wks	2 y.o.	^$^ Negative impact on base deficit within the first hour of birth
Vinnars MT et al. (2015) [39]	Chronic villitis	Delivered at 22–26 wks	2.5 y.o.	No case
Mir IN et al. (2020) [40]	Chronic villitis	Delivered at <29 wks	2 y.o.	No specific association, although non-specific multiple pathological lesions had negative association with neurodevelopmental impairment and/or bronchopulmonary dysplasia
Roescher AM et al. (2011) [41]	Chronic villitis of unknown origin, chronic villitis	Singleton pregnancy. Delivered at 25.4–31.7 wks	24 h	No association
van Vliet EOG et al. (2012) [42]	N/A	Singleton pregnancy. Delivered at <32 wks or birthweight of <1500 g	2 y.o./7 y.o.	No association
Gardella B et al. (2021) [26]	Chronic villitis, either of unknownsignificance or infectious villitis	Singleton pregnancy. FGR delivered at ≤34 wks and birthweight of ≤1500 g	2 y.o.	No association

wks: weeks of gestation. y.o.: years old. m.o.: months old. AD: appropriate description for APWGCS. SGA: small for gestational age. ^$^: MVM, FVM, and VUE were mixed together as a group of chronic placental abnormalities.

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
