# Peer review of "Association of Placental Pathology with Physical and Neuronal Development of Infants: A Narrative Review and Reclassification of the Literature by the Consensus Statement of the Amsterdam Placental Workshop Group"

_nutrients, 2024, doi:10.3390/nu16111786_

Round 1
Reviewer 1 Report
Comments and Suggestions for Authors
The aim of the authors is to reanalyse the scientific literature in the field of feto-placental pathophysiology, in light of The Consensus Statement of the Amsterdam Placental Workshop Group in order to evaluate the effects of placental pathology on the physical and intellectual development of infants
The topic is certainly current and of great interest, as well as necessary since there is still no coherent and standardized diagnostic and pathophysiological evaluation front in the international community that deals with fetoplacental pathology.
For this reason, the presentation in tables of the comparison between the various scientific publications in the field is very interesting This new discipline of neuroscience seeks to investigate the function of the placenta in neurodevelopmental diseases
Unfortunately the authors, in my opinion, do not consider a recent (from 2019) important scientific acquisition regarding the branch of neuroplacentology started by Professor Anna Penn. This new discipline of neuroscience seeks to investigate the function of the placenta in neurodevelopmental diseases and describes the dynamics of brain sparing in conditions of maternal-feto-placental malperfusion. I advise the authors to review the published articles, also citing a more recent article by Gardenella from 2022, rather than the one they cited from 2021.
We consider these minor points to be corrected:
On line 48, the article with reference 12 is too dated (2012), consider new evidence
Again on line 48 the authors state that they have demonstrated "We recently demonstrated that the fetal/placental weight ratio predicted the incidence of atopic dermatitis in female infants [13]" This is too strong an assertion, even for the impact factor of the journal , better to say that a close correlation has been hypothesized between fetoplacental pathology and dermatitis
As I said above, consider the reference of "Gardella B, Dominoni M, Scatigno AL, Cesari S, Fiandrino G, Orcesi S, Spinillo A. What is known about neuroplacentology in fetal growth restriction and in preterm infants: A narrative review of literature . Front Endocrinol (Lausanne). PMID: 36060976.
Mullen Scales of Early Learning (MSEL) composite scores, has not been adequately described
The data reported in line 273 and in table 6 are in contrast with what is well described and demonstrated for example by the article "Chalak L, Redline RW, Goodman AM, Juul SE, Chang T, Yanowitz TD, Maitre N, Mayock DE, Lampland AL, Bendel-Stenzel E, Riley D, Mathur AM, Rao R, Van Meurs KP, Wu TW, Gonzalez FF, Flibotte J, Mietzsch U, Sokol GM, Ahmad KA, Baserga M, Weitkamp JH, Poindexter BB, Comstock BA , Wu YW. Acute and Chronic Placental Abnormalities in a Multicenter Cohort of Newborn Infants with Hypoxic-Ischemic Encephalopathy. J Pediatr. 2021 Jun 16 . PMID: 34144032.
So I consider the article to be revised according to major revision
Reviewer 2 Report
Comments and Suggestions for Authors
Esteemed Authors
I have reviewed your manuscript, entitled **"Association of Placental Pathology with Physical and Neuronal Development of Infants: A Narrative Review and Reclassification of the Literature by The Consensus Statement of the Amsterdam Placental Workshop Group**. The authors were able to disclose a significant part of neonatal health: the association of placental pathology with the infant's development using standardized criteria according to the Consensus Statement of the Amsterdam Placental Workshop Group.
I suggest the following recommendations to improve the paper scientific value:
1. **Introduction and Background**:
- The background is quite solid and well-developed in presenting the justification of the study; however, the authors should have explained more and elaborately why the criteria used in the APWGCS classification were better or more reliable than those of the earlier classifications. This section would be better placed if specific examples of discrepancies or limitations in the earlier studies were provided.
2.
Authors reported in line 52 "Historically, it has been applied to the evaluation of placental conditions, including malfunction, and is referred to as the ‘memory of a pregnancy". To improve the value of this sentence, I suggest citing a novel and related original article that explore the uterine artery blood volume flow and its realtion with the placenta and birth weight:
- DOI: 10.1055/a-2075-3021
3. **
The methodology section is quite detailed, with explanations of the search strategy and criteria for the inclusion and exclusion of studies. However, a bit more detail on the process of reclassification would have been useful—how, for example, did the apparent problems with reclassification present themselves and how were these dealt with to ensure process consistency and accuracy?.
4. **
The reclassified results are well integrated into the discussion with existing literature. This could go a bit deeper into the clinical implications—how such findings affect or influence current prenatal care practices, or how current prenatal care practices could monitor high-risk pregnancies better.
Describe the limitations of the study a little more. Even though the manuscript states that a further study is needed, it could be more balanced to point out some methodological limitations, such as a possible selection bias on the studies or variations on the histopathological techniques.
5. ** Conclusion a bit short; more assertiveness could be applied to recommendations for further research. Make greater effort to identify where specific designated further research is required (e.g. the absence of any longitudinal studies on a low-risk population).
6. **Terminology and - Ensure the consistency of terms used all throughout the document. E.g. "maternal vascular malperfusion," "fetal inflammatory response" are some terms that usually cause confusion, if not used consistently.
Conclusion Your work provides essential information about the association between placental pathology and infant development. The major revisions are necessary to clarify and impact the manuscript. I look forward to seeing the revision.
Comments on the Quality of English Languageminor editing are needed
Round 2
Reviewer 1 Report
Comments and Suggestions for Authors
I now consider the authors' research work publishable